# Different Pulp Dressing Materials for the Pulpotomy of Primary Teeth: A Systematic Review of the Literature

**DOI:** 10.3390/jcm9030838

**Published:** 2020-03-19

**Authors:** Maurizio Bossù, Flavia Iaculli, Gianni Di Giorgio, Alessandro Salucci, Antonella Polimeni, Stefano Di Carlo

**Affiliations:** 1Department of Oral and Maxillofacial Science, “Sapienza” University of Rome, 00185 Rome, Italy; maurizio.bossu@uniroma1.it (M.B.); alessandro.salucci@uniroma1.it (A.S.); antonella.polimeni@uniroma1.it (A.P.); stefano.dicarlo@uniroma1.it (S.D.C.); 2Pediatric Dentistry School, Department of Oral and Maxillofacial Science, “Sapienza” University of Rome, 00185 Rome, Italy; flavia.iaculli@uniroma1.it

**Keywords:** biodentine, calcium hydroxide, ferric sulphate, MTA, primary teeth, pulpotomy

## Abstract

**Background**: Pulpotomy of primary teeth provides favorable clinical results over time; however, to date, there is still not a consensus on an ideal pulp dressing material. Therefore, the aim of the present systematic review was to compare pulpotomy agents to establish a preferred material to use. **Methods**: After raising a PICO question, the PRISMA guideline was adopted to carry out an electronic search through the MEDLINE database to identify comparative studies on several pulp dressing agents, published up to October 2019. **Results**: The search resulted in 4274 records; after exclusion, a total of 41 papers were included in the present review. Mineral trioxide aggregate (MTA), Biodentine and ferric sulphate yielded good clinical results over time and might be safely used in the pulpotomies of primary molars. Among agents, MTA seemed to be the material of choice. On the contrary, calcium hydroxide showed the worst clinical performance. Although clinically successful, formocreosol should be replaced by other materials, due to its potential cytotoxicity and carcinogenicity. **Conclusion**: MTA seemed to be the gold standard material in the pulpotomy of primary teeth. Promising results were also provided by calcium silicate-based cements. Further randomized clinical trials (RCTs) with adequate sample sizes and long follow-ups are encouraged to support these outcomes.

## 1. Introduction

Dental caries is an infective, chronic, degenerative and multifactorial condition that represents the most prevalent chronic disease worldwide, mainly in children [1,2]. Tooth decay would seem to be one of the major public health problems related not only to primary teeth but also to permanent ones, and, despite the preventive strategies mostly adopted in developed countries, 2.4 billion adults and 486 million children are affected by dental decay in the permanent and deciduous dentition, respectively [3].

Early caries management should avoid the progressive destruction of dental hard tissue and subsequent loss of dental vitality [4], inducing critical conditions in which premature tooth extraction is required [5]. This is mostly true for primary teeth (due to anatomical considerations, reduced rate of mineralization and high prevalence of risk factors) that show a rapid progression of tooth decay [2,4,6]. Therefore, vital pulp therapy (VPT) has been proposed to preserve the pulp vitality of deciduous or young permanent teeth with immature roots affected by caries and without evidence of radicular pathology [7,8]. Nowadays, treatment options of VPT are represented by indirect pulp treatment (namely indirect pulp capping), direct pulp capping and pulpotomy [7]. Although clinically successful in primary molars, direct capping is mainly recommended in the VPT of permanent young teeth [9,10] and indirect capping seems to possess a relative effectiveness when compared to pulpotomy procedures [11]. The latter provides favorable clinical survival rates over time and allows the vitality of primary teeth until their natural exfoliation, avoiding pulpectomy procedures [2]. Pulpotomy consists of elimination of the bacterial infection by the removal of the pulp in the pulp chamber; then, the decontaminated tooth is filled with a medicament [11]. The most frequently used agents are mineral trioxide aggregate (MTA), Biodentine (BD), formocresol (FC), ferric sulphate (FS) and calcium hydroxide (CH). When compared, FC, FS and MTA seemed to provide significantly better clinical and radiographic results as pulpotomy agents than CH after two years of follow-up; moreover, MTA showed the best performance in respect to FC and FS over time [12]. Accordingly, Stringhini et al. [13] reported that MTA yielded superior clinical and radiographical results in comparison to FC. On the other hand, electrosurgery and FS showed similar success to FC, whereas CH did not show positive evidence as medicament in pulpotomies of primary teeth [13]. In the same way, Asgary et al. [14] further stressed that MTA demonstrated better long-term outcomes in pulpotomy of primary molars when compared with FS.

More recently, bioactive endodontic cements have been introduced as valid alternatives to MTA in VPT, showing promising clinical results [15]. In addition, calcium-silicate-based cement demonstrates no difference when compared to MTA in the pulpotomies of primary teeth [4]; however, further long-term studies with larger sample sizes are needed to confirm these preliminary outcomes. 

To date there is still not an ideal pulp dressing material to be used in the pulpotomy of primary teeth. Therefore, the aim of the present systematic review was to compare several pulpotomy agents in order to establish a preferred material that performs better than others.

## 2. Materials and Methods

The present systematic review was conducted according to the PRISMA guidelines for Systematic Reviews [16]. The focused question was structured according to the PICO format (Population, Intervention, Comparison, and Outcome): is there a preferred material that performs better than others when used in pulpotomy of vital carious-exposed primary molars?

Population: Children with extensive caries involving vital dental pulp in primary teeth.

Intervention: Pulpotomy performed using different materials (MTA, Biodentine, ferric sulphate, calcium hydroxide).

Comparison: Between different materials applied in the same clinical conditions.

Outcome: Success of the therapy after at least 12 months of follow-up.

### 2.1. Search Strategy

An electronic search was conducted through the MEDLINE (PubMed) database to identify publications that met the inclusion criteria. The search was performed up to October 2019 in order to identify the studies that compare the performance of different materials in pulpotomy treatment of primary teeth, using the following search terms and key words alone or in combination with the Boolean operator “AND”: endodontics, pulpotomy, primary molars, deciduous teeth, primary teeth, biomaterials, biodentine, MTA, mineral trioxide aggregate, ferric sulphate, ferric sulfate, calcium hydroxide. Moreover, references of the eligible studies and relevant systematic reviews on the topic were manually checked and screened.

### 2.2. Study Selection

Two independent operators (F.I., G.D.G.) screened the studies according to the following inclusion/exclusion criteria:

#### 2.2.1. Inclusion Criteria

-Human in vivo studies written in English published in peer-reviewed journals;-Comparative clinical articles reporting on different materials applied in pulpotomy of primary teeth;-Definitive restorations of the primary teeth;-Clinical and/or radiographical follow-up of at least 12 months;-Random allocation of the samples.

#### 2.2.2. Exclusion Criteria

-In vitro studies on human and animals;-Systematic reviews, case series, case studies, retrospective studies;-Follow-up < 12 months;-Clinical studies without random allocation of the samples;-Non-comparative papers, namely reporting on only one material used in pulpotomy procedures;-Papers evaluating other clinical procedures that involved the pulp, such as direct capping, indirect capping, endodontic treatment.

After removing the duplicates, some papers were excluded subsequent to reading of the titles. Two review authors (F.I., G.D.G.) independently screened the selected abstracts to identify relevant studies according to the inclusion/exclusion criteria. In case of disagreement, a Senior Author (M.B.) was consulted and agreement was reached. Then, full reports of the selected studies were retrieved and a data extraction form was completed for each paper in an unblinded standardized manner, to determine whether the article should be included or excluded. Excluded studies and reasons for exclusion were reported.

### 2.3. Data Collection

Data extraction was performed by filling a form in with the following data: authors, title, publication year, aim, group distribution, materials compared, intervention, evaluated outcomes, reported results and conclusions.

After a preliminary evaluation of the selected papers, considerable heterogeneity was found in the study design, adopted procedures, outcome variables and results. Therefore, a descriptive analysis of the data was performed, since quantitative assessment and following meta-analysis could not be conducted. 

### 2.4. Assessment of Heterogeneity

The following variables were checked to determine heterogeneity:Pulpotomy procedureMaterials managementExpertise of the clinicianRestoration materialsOutcome variables

### 2.5. Quality Assessment

The assessment of methodological study quality was performed by two independent authors (F.I. and G.D.G.) following the recommendations for systematic reviews of interventions of the Cochrane collaboration [17] focusing on the following criteria: random sequence generation and allocation concealment (both accounting for selection bias), blinding of participants and personnel (performance bias), blinding of outcome assessment (detection bias), incomplete outcome data (attrition bias), selective reporting (reporting bias), or other possible causes of bias. 

Assessment of overall risk of bias was classified as follows: low risk of bias if all criteria were met; unclear risk of bias if one or more criteria were assessed as unclear; or high risk of bias if one or more criteria were not met [2].

## 3. Results

### 3.1. Search and Selection

The PubMed-MEDLINE search resulted in 4274 records. After duplicate removal, the titles and abstracts were screened according to the inclusion/exclusion criteria and a total of 75 papers underwent full-text reading. Thirty-four articles were excluded [18,19,20,21,22,23,24,25,26,27,28,29,30,31,32,33,34,35,36,37,38,39,40,41,42,43,44,45,46,47,48,49,50,51] since they did not meet the inclusion criteria; reasons of exclusion have been reported within Table 1. A total of 41 papers [52,53,54,55,56,57,58,59,60,61,62,63,64,65,66,67,68,69,70,71,72,73,74,75,76,77,78,79,80,81,82,83,84,85,86,87,88,89,90,91,92] were included in the present systematic review and processed for quality assessment and data extraction. The search strategy has been reported in Figure 1.

### 3.2. Assessment of Heterogeneity

The data extraction of the included studies yielded a considerable heterogeneity between the papers in terms of pulpotomy procedure, materials management, expertise of the clinician, restoration materials, and outcome variables. To better standardize the study comparison, papers reporting pulpotomy procedures different from the standard method were excluded (e.g., absence of the rubber dam, pulpotomy performed with laser ablation or electrosurgery, hemostasis obtained with several agents that could act as bias on the clinical outcomes).

Concerning materials management, the included studies evaluated several materials (e.g., MTA, BD, FS, CH, FC) that were applied with almost with the same procedure according to the manufacturer’s instructions; however, it should be considered that they were produced by various companies and might have a slightly different composition. Accordingly, the restoration materials reported by the included studies were different (composite, amalgam, glass ionomer cement, stainless steel crowns), however, in order to avoid bias, papers reporting teeth restored with temporary materials were excluded. Regarding the evaluated outcomes, all of the included studies assessed clinical and radiographical parameters; the success criteria used among the articles were similar but not the same and, therefore, it was only possible to make a descriptive comparison between the papers. Finally, the clinician expertise could not be evaluated in each study and the follow-up range varied between 12 and 42 months. Therefore, due to the lack of unequivocal data presentation, the results of the studies were reported separately.

### 3.3. Quality Assessment 

Assessments of the risk of bias and of the methodological study quality have been reported in Table 2. Overall risk of bias of the included studies showed high risk mainly in blinding of participants and personnel (28/41 studies), followed by blinding of outcome assessment (12/41 studies) (Figure 2). The lack of blind clinicians involved in the treatment as well as evaluation of the outcomes could affect the interpretation of the reported results provided in each study, playing a central role in the variability of study conclusions. 

The inter- inter-examiner agreement between the two independent authors that performed the quality assessment of the included studies was 0.95.

### 3.4. Outcomes

Data and results reported by each of the included studies are summarized in Table 3.

In order to ease the reading of the outcomes, the papers were further presented according to the material that yielded the best result after comparison.

#### 3.4.1. MTA

Almost 65% of the included papers (27/41) demonstrated that MTA provided comparable or even better results over time when compared to other materials used in the pulpotomy procedures of deciduous teeth. Specifically, MTA showed better performance than FC after 12 months of evaluation [67,76,86], with a statistically significant difference reported in two out three of the evaluated studies [67,86]. Moreover, better results of MTA in comparison to FC were observed after 24 months of follow-up [55,70,72,85,90], although the differences did not reach a statistical significance except in one study [55]. The same trend was maintained even after 30 [77] and 42 [78] months of evaluation, respectively. In two additional studies [66,88], it was reported that FC showed slightly worse results than MTA at a 24-month evaluation; however, it performed better than other materials assessed during pulpotomy of primary teeth, such as Pulpotec and Emdogain [88], as well as Portland cement and enamel matrix protein [66]. On the other hand, Jamali et al. [65] reported a superiority of MTA in respect to FC after 24 months of evaluation, even though both groups yielded worse results when compared to 3Mixtatin (a combination of simvastatin and 3Mix antibiotic) (78.9% for FC, 90.5% for 3Mixtatin and 88.1% for MTA). However, the differences between groups were not statistically significant.

When solely compared to BD, MTA showed slightly better performances after 12 [56], 18 [61] and 24 [52] months of assessment, without any statistically significant differences among groups. No differences between MTA and BD were reported by Juneja et al. [59], evaluating pulpotomy procedures on primary teeth performed also with FC. However, the authors observed that there were statistically significant differences between FC and MTA at 12 and 18 months, both clinically and radiographically, and between FC and BD at 12 and 18 months, only clinically [59]. Accordingly, Guven et al. [57] demonstrated no differences between BD and MTA groups (total success rates at 24 months were 82.75% BD, 86.2% MTA-P and 93.1% PR-MTA); however, in the same study, primary teeth treated with FS showed the lowest success rate (75.86%) at a 24-month follow-up, although this was not statistically significant. 

The comparison between MTA and FS yielded not significant differences after 18 [64] and 24 [71] months of evaluation; however, Doyle et al. [73] demonstrated a significantly lower survival rate for primary teeth treated with eugenol-free FS than MTA, after a follow-up period of 38 months. It should be noticed that Erdem et al. [71] not only reported the same performance for FS and FC (success rate of 88% for both groups) at a 24-month follow-up, but also demonstrated a statistically significant difference between MTA and a group of samples that underwent pulpotomy without use of any pulp dressing agent (96% vs. 68% after 24 months), suggesting the importance of the traditional pulpotomy procedure for the VTP of primary molars.

CH seemed to be the most ineffective material for pulpotomies of deciduous teeth and demonstrated the worst results when compared with MTA [63] after 12 months, and with MTA and FC (MTA 100%, FC 100%, CH 64%) [74], ProRoot MTA and MTA Angelus [68] and MTA and Portland cement [69] after 24 months of evaluation, respectively. In addition, the differences between CH and all tested materials were significantly different at all follow-up points.

Finally, the comparison of MTA with other pulpotomy agents, such as calcium-enriched mixture cement (CEM) [53] and Portland cement [54], provided the same clinical and radiographical performances of all evaluated materials after a follow-up period of 24 months.

#### 3.4.2. Biodentine

El Meligy et al. [87] clinically and radiographically evaluated 108 primary teeth that underwent pulpotomy performed with BD or FC. After 12 months, the authors reported a 100% clinical success rate in both groups and a radiographic success rate of 100% and 98.1% in the BD and FC groups, respectively, although without any statistically significant difference.

Three out of the 41 included papers reported the same [60] or even slightly better results [58,62] of BD in respect to MTA. Specifically, after a follow-up period of 12 months, 39 pulpotomized primary teeth treated with MTA showed a clinical success rate of 92% (36/39) and a radiographical success rate of 97% (38/39), whereas 39 teeth belonging to the BD group showed a clinical and radiographical success rate of 97% (38/39) and 95% (37/39), respectively [62]. A 24-month follow-up evaluation revealed that the clinical success rate of 62 primary molars that underwent pulpotomy was 96.8% (30/31) for both BD and MTA groups and the radiographic success was 93.6% (29/31) for the BD group and 87.1% (27/31) for the MTA group [58].

Therefore, although BD showed slightly better clinical results after one year [62] and radiographic results after two years of follow-up [58], no statistically significant differences were found among groups.

#### 3.4.3. Ferric Sulphate

A total of three out 41 included papers [80,83,92] demonstrated that FS performed better when compared to FC in the pulpotomy of carious deciduous teeth, however without reporting statistically significant differences. Specifically, after 12 months, a total success rate of 92.7% and 83.8% was reported by Fucks et al. [92] and a clinical success rate of 96.7% and 86.7% was reported by Havale et al. [80] in primary molars that underwent pulpotomy with FS and FC, respectively. The latest study [80] also demonstrated a gradual decrease of radiological success rate over time, showing rates of 56.7% and 63.3% for FS and FC, respectively. Moreover, Ozmen et al. [83] compared three pulpotomy agents, such as FC, FS and Ankaferd blood stopper (ABS), and reported a more favorable clinical success rate for FS (100%) than other evaluated materials (87% for both ABS and FC) after a follow-up of 24 months. Concerning radiographical success, the same authors reported gradually reduced rates that were comparable for FS and ABS (87%) and slightly lower for FC (80%).

#### 3.4.4. Formocresol

According to the International Agency for Cancer Research, one of the main components of FC, namely formaldehyde, has been classified as a human carcinogen [93]; due to this reason, FC was not included as one of the keywords in the search strategy of the present systematic review. However, the same material is still largely used and was reported in more than half of the included studies (23/41). Among them, seven papers [75,79,81,82,84,89,91] reported similar or even better results of FC when compared to other agents used in pulpotomy of primary teeth. Durmus et al. [79] reported a 12-month clinical success rate of 97% and 92.5% of deciduous teeth pulpotomized and treated with FC and FS, respectively, as well as comparable radiographical results (87% FC vs.79% FS), without any statistically significant differences among groups. Moreover, FC and FS provided similar results in pulpotomy procedures after 12 (clinical success: 96% FC and 95.7% FS; radiographic success: 100% both FC and FS) and 18 months (clinical success: 96% FC and 87% FS; radiographic success: 100% FC and 91.3% FS) of evaluation [84]. Markovic et al. [82] compared the 18-month clinical and radiographical success of pulpotomies performed on 104 primary molars randomly divided into three groups and treated with FS, FC and CH. FS and FC showed comparable radiographical and clinical success (89.2% and 90.9%, respectively); on the other hand, the CH group demonstrated lower success than other groups (82.3%), although this was not statistically significant [82]. Accordingly, comparing pulpotomies with FS, FC and CH after 12, 24 and 36 months, CH showed the worst results after 24 and 36 months and, even though the values did not reach statistical significance, the failure rate for the CH group was three times higher than the FC one [81]. On the other hand, primary teeth treated with FC after pulpotomy showed slightly better results than the FS group after 12 months of evaluation (96% FC vs. 86% FS), and vice versa after 24 and 36 months of follow-up (85% FC vs. 86% FS and 72% FC vs. 76% FS, respectively) [81]. Fernandes et al. [89] reported a significantly better radiographical success rate of pulpotomy performed with FC compared to CH after 12 (100% FC vs. 50% CH) and 18 months (100% FC vs. 66.7% CH), demonstrating that CH may not be considered suitable in pulpotomy treatment of primary molars, even in combination with Low Level Laser Therapy [89]. Similar outcomes were also reported by Sonmez et al. [91], who observed 2-year follow-up success rates of 46.1%, 66.6%, 73.3% and 76.9% in 80 primary molars treated with CH, MTA, FS and FC, respectively. Although no statistically significant differences were detected among groups, CH seemed to be less clinically appropriate than other evaluated materials. Finally, Noorollahian [75] reported that, after 24 months of evaluation, primary teeth treated with FC during pulpotomy provided better radiographical results than ones that underwent MTA, although both groups yielded a 100% clinical success at the same follow-up point.

## 4. Discussion

VTP aims at preserving pulpal tissue and promoting repair of the mineralized tissue barrier (dentin bridge) [94]. In addition, the success of this technique would avoid pulpectomy and subsequent root canal obturation by several materials, that, on turn, could prevent the radicular resorption of the primary molars and alter the development of the permanent teeth [11].

Since there is a lack of a general consensus regarding an ideal pulp dressing material, the aim of the present systematic review was to establish a preferred agent to be used in the pulpotomy procedure of primary teeth affected by deep caries, after raising a PICO question. The evaluation of the included studies suggested that MTA seemed to be the material of choice after pulpotomies. Although it showed successful clinical performances over time, the majority of the authors agreed on its drawbacks, such as high costs, difficult storage and long setting time [4]. Therefore, in some cases, alternative materials may be used. FC had historically been indicated as a valid option in the pulpotomy procedures of primary molars; however, the evidence-based scientific literature has already demonstrated its potential cytotoxicity and carcinogenicity [93]. Due to this reason, FC was not included in the search strategy of the present systematic review; nevertheless, it is largely used and provides some good clinical results. Thus, to supply a complete overview on the topic, papers that compared several materials with FC were included. Seven studies [75,79,81,82,84,89,91] reported better clinical outcomes of FC than FS. On the other hand, the comparison between FC and MTA [55,66,67,70,72,76,77,78,85,86,88,90], yielded a better performance of the latter after 12, 24, 30 and 42 months of evaluation. Accordingly, El Meligy et al. [87] observed slightly favorable clinical and radiographical outcomes of primary teeth underwent pulpotomy performed with BD than FC, although no statistically significant.

FS yielded more favorable clinical results when compared to FC in 3/41 studies included in the present review [80,83,92]. Even though it provided comparable or slightly worse outcomes than MTA [64,71,73], when the pulpotomized primary molars are going to be replaced by permanent teeth, FS may be used as a safe alternative [95].

In accordance with the scientific literature [95], the present review confirmed that CH seemed to be the most ineffective material for pulpotomies of deciduous teeth and demonstrated the worst results when compared with all tested materials, reaching statistically significant differences at all follow-up points [52,63,69,74,81,89].

The introduction of calcium-silicate-based cements (such as Biodentine) appears to be promising for VTP. Indeed, calcium-silicate-based cements seem to play a central role in regenerative endodontics, inducing pulp regeneration, healing and dentin formation [96]. The present review confirms the previously reported results [4,15], showing similar outcomes when MTA was compared to BD [52,56,57,58,59,60,61,62]. MTA and BD may be classified as bioactive endodontic cements, due to their bioactivity feature, despite the differences in their chemical compositions [15]. The encouraging clinical properties as well as biocompatibility of calcium-silicate-based cements indicate that they can be considered as a suitable alternative to MTA for pulpotomies in primary molars. However, these preliminary results should be supported by further studies.

### Limitations

The main limitation of the present systematic review was the high heterogeneity of the included studies. Although only randomized clinical comparative studies with at least 12 months of follow-up were evaluated, the lack of univocal standard procedures made difficult a precise comparison of the data. Moreover, the use of several materials composition as well as slightly different outcomes evaluation provided high variability in the interpretation of the results and could let to a misjudgment in the Conclusions. Due to this reason, some “confounding” materials reported by several included studies, such as sodium hypochlorite [84], Er:YAG laser [81], diode laser [79] and low level laser therapy [89], were excluded in the evaluation of pulpotomy dressing agents.

It should be further considered the high variability given by the type of restoration material used, although definitive, its interaction with the pulpotomy agent as well as the inconstant time between the pulpotomy treatment and the physiological exfoliation of the same tooth, that would render very hard to establish the success of pulpotomy procedure over time.

The quality assessment of the included studies showed an overall high risk of bias, mainly in blinding of participants and personnel, followed by blinding of outcome assessment. This aspect highlighted the inadequacies in the published studies, as previously reported by Gopalakrishnan et al. [97]. High quality study design and standardized clinical and radiographical protocols are needed to prospectively assess the performances of pulpotomy medicaments used in deciduous teeth.

## 5. Conclusions

Within the limitation of the present systematic review, MTA seemed to be the gold standard material in the pulpotomy of primary teeth. Promising results were also provided by BD. On the contrary, CH should be firmly avoided during pulpotomy procedures. Further RCT studies with adequate sample sizes and long follow-ups are encouraged to confirm these outcomes.

## Figures and Tables

**Figure 1 jcm-09-00838-f001:**
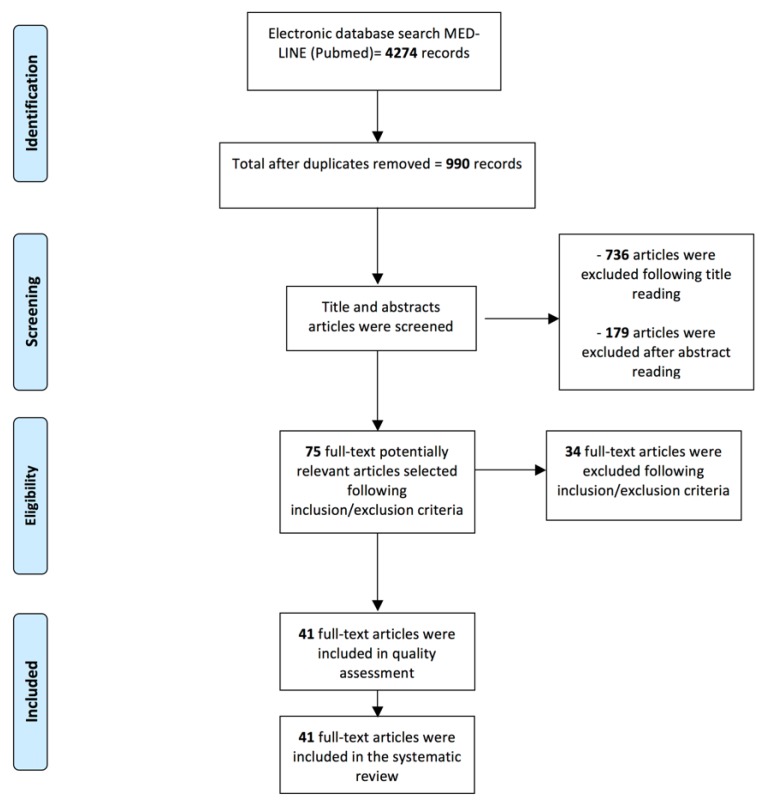
Flowchart of the review process and search strategy according to PRISMA statement.

**Figure 2 jcm-09-00838-f002:**
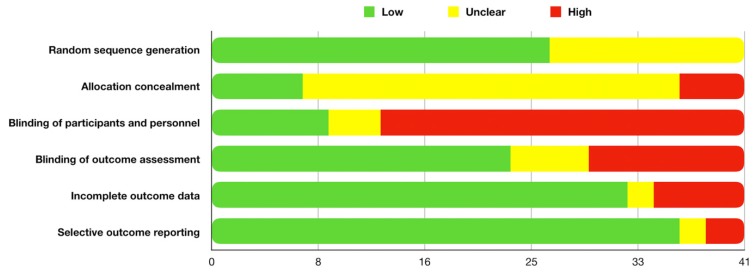
Overall risk of bias.

**Table 1 jcm-09-00838-t001:** Excluded studies and reason of exclusion.

Author, Year	Reason of Exclusion
Kathal et al. 2017 [18]	The studied material did not present clinical evidence among scientific literature.
Alsanouni et al. 2019 [19]	Authors compared the same pulpotomy dressing material.
Pratima et al. 2018 [20]	Pulpotomy was performed by diode laser prior to MTA.
Kang et al. 2015 [21]	Authors compared the same pulpotomy dressing material.
Akcay et al. 2014 [22]	Sodium hypochlorite was applied prior to MTA and might act as a variable.
Fernández et al. 2013 [23]	Internal root resorption was not considered as a failure.
Liu et al. 2011 [24]	Calcium hydroxide paste was mixed with other agents and the obtained material did not present clinical evidence among scientific literature.
Holan et al. 2005 [25]	Internal root resorption was not considered as a failure.
Nematollahi 2018 [26]	Authors performed partial pulpotomy that is poorly reproducible and standardizable.
Musale et al. 2016 [27]	The studied material did not present clinical evidence among scientific literature.
Atasever et al. 2019 [28]	Sodium hypochlorite was used during pulpotomy procedure and might act as a variable.
Huth et al. 2005 [29]	The paper reported on the same sample size of Huth et al. 2012.
Nguyen et al. 2017 [30]	Pulpotomy was compared with root canal therapy.
Saltzman et al. 2005 [31]	Pulpotomy procedures were different between the evaluated groups.
Grewal et al. 2016 [32]	The success of the materials was evaluated on dentin thickness without reproducibility and standardization.
Hugar et al. 2017 [33]	Incomplete data reported.
Kalra et al. 2017 [34]	The studied material did not present clinical evidence among scientific literature.
Uloopi et al. 2016 [35]	Pulpotomy procedures were different between the evaluated groups.
Yildiz et al. 2014 [36]	No random allocation of the sample size.
Ansari et al. 2018 [37]	Absence of rubber dam.
Gupta et al. 2015 [38]	Pulpotomy procedures were performed by laser or electrosurgery.
Cantekin et al. 2014 [39]	Authors compared the same pulpotomy dressing material.
Trairatvorakul et al. 2012 [40]	Authors performed partial pulpotomy that is poorly reproducible and standardizable.
Zurn et al. 2008 [41]	Pulpotomy was obtained by light-cured calcium hydroxide.
Percinoto et al. 2006 [42]	Corticosteroid/antibiotic solution was applied as therapeutic dressing and might act as a variable.
Ghoniem et al. 2018 [43]	No random allocation of the sample size.
Biedm-Perea et al. 2017 [44]	Retrospective study and no random allocation of the sample size.
Airen et al. 2012 [45]	Retrospective study and no random allocation of the sample size.
Frenkel et al. 2012 [46]	No random allocation of the sample size.
Cardoso Silva et al. 2011 [47]	No random allocation of the sample size.
Ibricevic et al. 2003 [48]	Retrospective study.
Godhi et al. 2011 [49]	No random allocation of the sample size.
Hugar et al. 2010 [50]	No random allocation of the sample size.
Ibricevic et al. 2000 [51]	No random allocation of the sample size.

**Table 2 jcm-09-00838-t002:** Assessment of risk of bias of the included studies.

	Random Sequence Generation	Allocation Concealment	Blinding of Participants and Personnel	Blinding of Outcome Assessment	Incomplete Outcome Data	Selective Outcome Reporting
Çelik et al. 2019 [52]	Low	Unclear	High	Low	Low	Low
Malekafzali et al. 2011 [53]	Unclear	High	High	Low	Unclear	Unclear
Sakai et al. 2009 [54]	Low	High	High	Low	High	High
Farsi et al. 2005 [55]	Low	Unclear	Unclear	Unclear	High	High
Carti et al. 2017 [56]	Low	High	High	Unclear	Low	Low
Guven et al. 2017 [57]	Low	High	Low	Low	Low	Low
Bani et al. 2017 [58]	Low	Unclear	Unclear	High	Low	Low
Juneja et al. 2017 [59]	Low	Unclear	Unclear	Low	Low	Low
Togaru et al. 2016 [60]	Unclear	High	High	High	Low	Low
Rajasekharan et al. 2017 [61]	Low	Low	Low	Low	Low	Low
Cuadros-Fernández et al. 2016 [62]	Low	Unclear	High	High	Low	Low
Silva et al. 2019 [63]	Low	Low	Low	Low	Low	Low
Junqueira et al. 2018 [64]	Low	Unclear	High	Low	High	Low
Jamali et al. 2018 [65]	Low	Unclear	Low	Low	High	Low
Yildirim et al. 2016 [66]	Unclear	Unclear	High	High	Low	Low
Olatosi et al. 2015 [67]	Unclear	Unclear	High	High	Low	Low
Celik et al. 2013 [68]	Low	Low	Low	Low	Low	Low
Oliveira et al. 2013 [69]	Low	Unclear	High	Low	Low	Low
Sushynski et al. 2012 [70]	Unclear	Unclear	High	Low	Low	Low
Erdem et al. 2011 [71]	Unclear	Unclear	Low	High	Low	Low
Ansari et al. 2010 [72]	Unclear	Unclear	High	Unclear	Low	Low
Doyle et al. 2010 [73]	Low	Low	Unclear	Low	Unclear	Low
Moretti et al. 2008 [74]	Low	Unclear	High	Low	Low	Low
Noorollahian 2008 [75]	Low	Unclear	High	Low	High	Low
Agamy et al. 2004 [76]	Unclear	Unclear	High	Low	Low	Low
Eidelman et al. 2001 [77]	Low	Unclear	High	High	High	Low
Mettlach et al. 2013 [78]	Low	Unclear	High	Low	High	Low
Durmus et al. 2014 [79]	Unclear	Unclear	High	Low	Low	Low
Havale et al. 2013 [80]	Unclear	Unclear	High	Unclear	Low	Low
Huth et al. 2012 [81]	Low	Low	Low	Low	Low	Low
Markovic et al. 2005 [82]	Unclear	Unclear	High	Unclear	Low	Low
Ozmen et al. 2017 [83]	Low	Unclear	High	High	Low	Low
Farsi et al. 2015 [84]	Low	Low	Low	Low	Low	Low
Jayam et al. 2014 [85]	Unclear	Unclear	High	High	Low	Low
Srinivasan et al. 2011 [86]	Unclear	Unclear	High	Low	Low	Low
El Meligy et al. 2019 [87]	Low	Low	Low	Low	Low	Low
Sunitha et al. 2017 [88]	Unclear	Unclear	High	High	Low	Low
Fernandes et al. 2015 [89]	Low	Unclear	High	Low	Low	High
Subramaniam et al. 2009 [90]	Low	Unclear	High	High	Low	Low
Sonmez et al. 2008 [91]	Unclear	Unclear	High	High	Low	Low
Fuks et al. 1997 [92]	Low	Unclear	High	Unclear	Low	Unclear

**Table 3 jcm-09-00838-t003:** Summary of the data reported in the studies included in the present systematic review.

	Material	Groups Distribution	Type of Definitive Restorations	Follow-up	Evaluated Outcomes		Reported Outcomes		Conclusions
					Clinical	Radiographical	Clinical	Radiographical	
Çelik et al. 2019 [52]	MTA***** vs. BD**°**	MTA group (*n* = 24) BD group (*n* = 20)	IRM and SCC	12, 18 and 24 months.	Absence of spontaneous pain and/ or sensitivity to palpation/percussion; absence of fistula, swelling, and/or abnormal mobility.	Absence of radiolucencies at the inter-radicular and/or periapical regions, absence of pulp canal obliteration (fully obliterated canals); absence of internal or external (pathologic) resorption that was not compatible with a normal exfoliation process.	MTA = 100% success rate at 12, 18 and 24 months. BD = 89.4% success rate at 12, 18 and 24 months.	MTA = 100% success rate at 12, 18 and 24 months. BD = 89.4% success rate at 12, 18 and 24 months.	MTA and BD showed similar success rates without any statistically significant difference.
Malekafzali et al. 2011 [53]	MTA* vs. CEM§	MTA group (*n* = 40) CEM group (*n* = 40)	SCC or amalgam depending on the cavity size	12 and 24 months	Swelling/abscess, sinus tract, spontaneous pain, and or pathological mobility.	Furcation radiolucency, periapical bone destruction, internal root resorption, and pathological external root resorption.	MTA = 100% success rate at 12, 18 and 24 months. CEM = 100% success rate at 12, 18 and 24 months.	One and three cases of pathologic external root resorption were observed in CEM and MTA groups at 12-month follow-up, respectively, without significant difference. In the last follow-up (24 months) MTA and CEM achieved 100% radiographic success.	The study demonstrated favorable treatment outcomes of CEM/MTA pulpotomy in human primary molar teeth. CEM as a new endodontic cement is a promising biomaterial.
Sakai et al. 2009 [54]	Grey MTA****** vs. PC**#**	MTA group (*n* = 15) PC group (*n* = 15)	IRM and GIC	12, 18 and 24 months	Absence of spontaneous pain, mobility, swelling, fistula, or smell.	Absence of internal root resorption or furcation radiolucency.	100% of the available teeth were clinically and radiographically successful during all the follow-ups.	100% of the available teeth were clinically and radiographically successful during all the follow-ups	The present data suggested that PC might serve as an effective and less expensive MTA substitute in primary molar pulpotomies.
Farsi et al. 2005 [55]	MTA vs. FC (both not specified)	MTA group (*n* = 60) FC group (*n* = 60)	IRM was placed prior to restoration with SCC.	12, 18 and 24 months	Absence of pain; swelling; sinus tract; mobility; or pain on percussion.	Absence of internal root resorption; furcation radiolucency; periapical radiolucency; or widening of the periodontal ligament space.	After 24 months, the FC group showed only one case reported pain. On the other hand, 100% of teeth treated with MTA were considered clinically successful.	At the end of the study, the FC group showed five cases with pulp pathosis (13.2%). MTA showed 100% of radiographical success.	MTA might be considered as a valid alternative to FC.
Carti et al. 2017 [56]	MTA (not specified) vs. BD**°**	MTA group (*n* = 25) BD group (*n* = 25)	- MTA group: GIC and SCC cemented with GIC. - BD group: the cavity was filled with BD and then restored by using a SCC cemented with GIC.	12 months	Absence of palpation–percussion sensitivity, spontaneous pain, hot–cold sensitivity, presence of fistula-swelling, pathologic mobility.	Absence of internal–external resorption, periapical/interradicular bone destruction, disintegration of the lamina dura, enlargement of the periodontal space, and radiological calcific metamorphosis.	There was no statistically significant difference between clinical success rates over time. In both groups one tooth was extracted due to fistula formation at month 12.	The success rates were 80% and 60% for MTA and BD groups, respectively. There were no statistically significant differences between the groups.	Both MTA and BD could be used as pulpotomy agents, but more long-term studies with larger sample sizes are required.
Guvenet al. 2017 [57]	MTA-P******* vs. PR-MTA***** vs. BD**°** vs. FS (not specified)	MTA-P group (*n* = 29) PR-MTA group (*n* = 29) BD group (*n* = 29) FS group (*n* = 29)	- MTA groups: GIC was placed over the MTA. - BD group: permanent restoration was performed on the same session with GIC. - FS group: a ZOE base, then GIC	12 and 24 months	Absence of swelling, pain, fistula, or pathologic mobility.	Absence of evidence of internal or external resorption or periradicular radiolucency.	24-month: no clinical failure was observed among groups. Total success rates of the BD, MTA-P, PR-MTA and FS groups were 82.75%, 86.2%, 93.1% and 75.86%, respectively. No statistically significant differences in total success rates were observed over time.	Overall, seven teeth demonstrated radiographic failure at 24 months.	This study found no statistically significant differences among pulpotomy techniques; however, calcium-silicate-based materials appeared to be clinically more appropriate than FS.
Bani et al. 2017 [58]	MTA***** vs. BD**°**	MTA group (*n* = 32) BD group (*n* = 32)	GIC and SCC	12, 18 and 24 months	Absence of tenderness to percussion, swelling, pain, fistula, or pathologic mobility.	Absence of internal or external resorption; furcal or periradicular radiolucency; widening of periodontal ligament spaces.	The 24-month follow-up evaluations revealed that the clinical success rates were 96.8% for both BD and MTA groups.	The radiographic success rates at 24 months were 93.6% for BD and 87.1% for MTA.	BD and MTA did not differ significantly in combined clinical and radiographic success after 24 months. However, BD showed slightly better radiographical results after two years of follow-up.
Juneja et al. 2017 [59]	MTA***** vs. BD**°** vs. FC (not specified)	MTA group (*n* = 17) BD group (*n* = 17) FC group (*n* = 17)	All teeth were immediately restored with IRM and GIC, then were restored with pre-formed metal crowns.	12 and 18 months	Absence of pain, tenderness to percussion/palpation, swelling, intraoral/extraoral sinus, pathologic mobility.	Absence of internal or external resorption; furcal or periradicular radiolucency.	100% of available teeth for MTA and BD groups were clinically successful, and 73.3% of the FC group. There were statistically significant differences between FC and MTA and BD at 12 and 18 months, respectively.	Radiographic success rate for the FC group at 18 months follow up was 73.3% for FC, 100% for MTA and 86.6% for BD group. There were statistically significant differences between FC and MTA at 12 and 18 months.	MTA and BD showed more favorable results than FC.
Togaru et al. 2016 [60]	MTA***** vs. BD**°**	MTA group (*n* = 45) BD group (*n* = 45)	Permanent restoration with GIC followed by SCC	12 months	Absence of pain, tenderness on percussion, swelling and/or fistula, pathologic tooth mobility.	Absence of radiolucency in furcation/periapical area, internal or external root resorption, and widening of periodontal space.	12 months: MTA and BD provided 95.5% of success rate.	12 months: MTA and BD provided 95.5% of success rate. Radiographic examination provided 1 failure in both MTA and BD groups. No statistical differences were detected.	Pulpotomy treatment using BD and MTA had similar success rates in primary teeth.
Rajasekharanet al. 2017 [61]	MTA***** vs. BD**°** vs. TP**##**	MTA group (*n* = 29) BD group (*n* = 25) TP group (*n* = 27)	GIC and SCC	12 and 18 months	Absence of pain, tenderness on percussion, swelling and/or fistula, pathologic tooth mobility, chewing sensitivity, gingival inflammation, periodontal pocket formation, sinus tract present, premature tooth loss due to pathology.	Absence of radiolucency in furcation/periapical area, internal or external root resorption, and widening of periodontal space, variation radiodensity.	Clinical success was 95.24%, 100% and 95.65% in the BD, MTA and TP groups, respectively.	Radiographic success was 94.4%, 90.9% and 82.4% in the BD, MTA and TP groups, respectively	After 18-month follow-up, there was no significant difference between BD in comparison with MTA or TP.
Cuadros-Fernández C et al. 2016 [62]	MTA***** vs. BD**°**	MTA group (*n* = 43) BD group (*n* = 41)	IRM and SCC.	12 months	Absence of pain, swelling or gingival inflammation, fistulation, or pathologic mobility.	Absence of evidence of internal or external resorption or periradicular radiolucency.	The clinical success rate in the MTA group after 12 months was 92% (36/39), whereas the clinical success rate in the BD group after 12 months was 97% (38/39).	MTA yielded a radiographic success of 97% (38/39). Use of BD yielded a radiographic success of 95% (37/39).	BD showed similar clinical results as MTA with comparable success rates when used for pulpotomies of primary molars.
Silva et al. 2019 [63]	MTA** (only gray), CH (not specified) with saline (CH+saline group) and CH with polyethylene glycol (CH+PEG group)	MTA group (*n* = 15) CH+saline group (*n* = 15) CH+PEG group (*n* = 15)	1-mm-thick layer of material was used for capping, followed by another 1-mm-thick of a layer of cement-cured CH°° employed as an intermediate base for the restoration GIC	12 months	Lack of spontaneous pain, mobility, swelling, or fistula in the treated tooth.	Lack of internal or external root resorption and furcation radiolucency were indicative of radiographic success.	Clinical analysis showed 100% treatment success using MTA, at all follow-up appointments.	Radiographic analysis showed 100% treatment success using MTA, at all follow-up appointments. At 12 months of follow-up, the CH+saline group had an increased incidence of radiographic failure compared with the MTA group.	The association of CH with PEG provided better results than that of CH + saline as a capping material for pulpotomy of primary teeth. However, both associations demonstrated clinical and radiographic results inferior to those of MTA.
Junqueira et al. 2018 [64]	MTA** vs. FS§§	MTA (*n* = 15) FS (*n* = 16)	IRM was placed prior to the restoration with GIC	12 and 18 months	Absence of spontaneous pain, mobility, swelling or fistula.	Absence of internal root resorption, inter-radicular radiolucency and periapical lesion were absent. Hard tissue barrier formation and stenosis were considered as radiographic successes; tooth discoloration was not considered as a failure.	In both groups, 100% of the available teeth were clinically successful during all the follow-up appointments.	The radiographic success rate for both groups was 100% at 12 months. At the end of the 18-month follow-up period, one tooth from FS group presented a radiographic failure (inter-radicular radiolucency), but it was not statistically different from MTA group.	Based on this study, both MTA and 15.5% FS are effective for pulpotomies of primary teeth. Although MTA is considered the first choice material, FS may be a suitable alternative when treatment cost is an issue.
Jamali et al. 2018 [65]	3Mixtatin vs. FC**°°°** vs. MTA******	3Mixtatin group (*n* = 50) FC group (*n* = 50) MTA group (*n* = 50)	IRM and amalgam	12 and 24 months	Absence of sinus tract, tenderness to palpation and percussion, spontaneous pain or pain of long duration, swelling, pain of other sources mimicking irreversible pulpitis such as a gingival problem, food impaction, etc.	Absence of external or internal root resorption, inter-radicular radiolucency and periapical lesion.	The overall success rate was 78.9% for FC, 90.5% for 3Mixtatin and 88.1% for MTA group. There was no significant difference in overall success rate among the groups after 24-month follow-up.	The overall success rate was 78.9% for FC, 90.5% for 3Mixtatin and 88.1% for MTA group. There was no significant difference in overall success rate among the groups after 24-month follow-up.	The present study showed that 3Mixtatin can be utilized as a pulp capping material in pulpotomy of primary teeth owing to its successful clinical and radiographic outcomes after 24 months of follow-up period.
Yildirim et al. 2016 [66]	FC**°°°** vs. MTA* vs. PC**#** vs. EMP	FC group (*n* = 35) MTA group (*n* = 35) PC group (*n* = 35) EMP group (*n* = 35)	GIC and SCC	12 and 24 months	Absence of spontaneous pain, swelling, fistula	Absence of radiolucency of the periapical or furcation, and pathological external root resorption, internal root resorption	24 months: FC = 96.9%, MTA = 100%, PC = 93.3%, EMD = 90.6%.	24 months: FC = 96.9%, MTA = 100%, PC = 93.3%, EMD = 90.6%.	This study demonstrated that MTA had better long-term clinical success rates than FC, PS and EMP, respectively.
Olatosi et al. 2015 [67]	FC§§§ vs. White MTA*	MTA group (*n* = 25) FC group (*n* = 25)	SSC	12 months	Absence of symptoms of pain, tenderness to percussion, swelling or sinus tract, pathologic tooth mobility.	Absence of periodontal ligament widened, furcation or periapical radiolucency, active/progressing internal root resorption, pathologic external root resorption.	The clinical success rate at 12 months was 100% and 81% for MTA and FC, respectively. The difference was statistically significant.	The radiographic success rates for MTA and FC were 96% and 81%, respectively. There was no statistically significant difference between the two agents.	MTA showed clinical and radiographic success as a dressing material following pulpotomy procedure in primary teeth, and it has a promising potential to become a replacement for FC in primary molars.
Celik et al. 2013 [68]	MTA*(P-MTA) vs. MTA** (A-MTA) vs. CH (not specified)	P-MTA group (*n* = 46) A-MTA group (*n* = 45) CH group (*n* = 48)	GIC and amalgam	12, 18 and 24 months	Absence of spontaneous pain, sensitivity to palpation/percussion, fistula, swelling, abnormal mobility.	Absence of radiolucencies at the inter-radicular and/or periapical regions, pulp canal obliteration (fully obliterated canals), internal or external resorption.	Comparisons using the log-rank test showed that the clinical survival probabilities of P-MTA and A-MTA were similar and significantly greater than that of the CH group, respectively.	The 24-month cumulative radiographic survival probabilities of the P-MTA, A-MTA, and CH groups were 0.974, 0.908, and 0.446, respectively. Most radiographic failures were associated with internal resorption, which was observed in 23 teeth in the CH group, compared to none in the P-MTA and three in the A-MTA groups.	Based on the results of this study, P-MTA and A-MTA showed high clinical and radiographic success rates as pulpotomy agents in primary molars. CH showed considerably less clinical and radiographic success than the MTA cements.
Oliveira et al. 2013 [69]	CH† vs. MTA****** vs. PC#	CH group (*n* = 15) MTA group (*n* = 15) PC group (*n* = 15).	IRM and GIC	12 and 24 months	Absence of spontaneous pain, mobility, swelling and fistula.	Absence of internal root resorption and furcation radiolucency.	Clinically, the MTA and PC groups showed 100 % success rates at 12 and 24 months.	Radiographically, the MTA and PC groups showed 100 % success rates at 12 and 24 months.	MTA and PC might serve as effective materials for pulpotomies of primary teeth as compared to CH. Although results are encouraging, further studies and longer follow-up assessments are needed in order to determine the safe clinical indication of Portland cement.
Sushynski et al. 2012 [70]	Gray MTA***** vs. DFC	MTA group (*n* = 119) DFC group (*n* = 133)	IRM and SSC	24 months	Absence of mobility, percussion or chewing sensitivity, gingival inflammation, pathology, periodontal pocket formation, spontaneous pain, sinus tract presence, premature tooth loss due to pathology.	Absence of internal root resorption (nonperforated/perforated); external root resorption; dentin bridge formation; pulp canal obliteration/calcific metamorphosis; furcal/periradicular radiolucencies, widening of the periodontal ligament space; periapical bone destruction; physiological root resorption.	All teeth in the MTA group were judged to be clinically successful (100%), whereas 1% of teeth in the DFC group were judged to have failed from 6 to 24 months (success ~99%). The differences between groups were not significant at all follow-up points.	At the 24-month follow-up 62/65 (~95%) molars of the MTA group were radiographically successful, while only 50/66 (~76%) molars of the DFC group demonstrated radiographic success.	MTA demonstrated significantly better radiographic outcomes vs. the DFC. However, both pulpal agents, presented comparable clinical outcomes after two years of follow-up.
Erdem et al. 2011 [71]	MTA* vs. FS§§ vs. DFC vs. ZOE	MTA group (*n* = 32) FS group (*n* = 32) FC group (*n* = 32) ZOE group (*n* = 32)	amalgam	12 and 24 months	Absence of spontaneous pain or after percussion, mobility, swelling.	Absence of internal root resorption and furcation and/or periapical bone destruction.	12 months success: 100% for MTA, FC and FS., and 92% for ZOE. 24 months success: 96% MTA, 88% FS, 88% FC and ZOE 68%.	12 months success: 100% for MTA, FC and FS., and 92% for ZOE. 24 months success: 96% MTA, 88% FS, 88% FC and ZOE 68%.	ZOE, as the only pulpotomy medicament, had a significantly lower success rate than MTA. No significant differences were observed, among the 3 experimental materials (MTA, FC and FS) at two years follow-up.
Ansari et al. 2010 [72]	MTA***** vs. DFC	MTA group (*n* = 20) FC group (*n* = 20)	SSC	12 and 24 months	Absence of pain, presence of gingival swelling and sinus tract.	Absence of internal resorption, radiographic signs of pathosis (periapical radiolucency).	The number of teeth judged as failed was six in the FC-treated group with only one failed case in the MTA-treated group	Overall radiographic success at 24th month was observed in > 95% of MTA group and 90% of FC group	Pulpotomy of primary teeth performed with MTA demonstrated comparable results of FC-treated teeth.
Doyle et al. 2010 [73]	MTA***** vs. FS§§ vs. Eugenol-free FS§§ vs. FS/MTA	FS group (*n* = 58) MTA group (*n* = 57) Eugenol-free FS group (*n* = 78) FS/MTA group (*n* = 77)	IRM and SSC	12, 24 and 36 months	Absence of SCC perforation, mobility, percussion sensitivity, palpation sensitivity, soft tissue pathology.	Absence of widening of the periodontal ligament space, furcal/periradicular radiolucencies, pulp canal obliteration, internal or external root resorption.	Eugenol-free FS molars demonstrated significantly lower survival rates than MTA ones, over 6 to 38 months.	MTA molars demonstrated significantly fewer radiographical changes than FS ones. Eugenol-free FS showed significantly more radiographical changes than MTA or FS/MTA.	MTA showed statistically significant better performances than FS and Eugenol-free FS
Moretti et al. 2008 [74]	MTA** vs. CH°° vs. DFC	MTA group (*n* = 15) CH group (*n* = 15) DFC group (*n* = 15)	IRM and GIC	12, 18 and 24 months	Absence of spontaneous pain, mobility, swelling, fistula and smell.	Absence of internal root resorption, inter-radicular bone destruction and furcation radiolucency.	Both groups showed100% of clinical success during all the follow-up appointments. The CH group demonstrated 64% of success.	Both groups showed 100% radiographical success during all the follow-up appointments. The CH group demonstrated 64% success; in the same group, internal resorption was a frequent radiographic finding.	MTA was superior to CH and equally effective to DFC as a pulpotomy agent in primary molars.
Noorollahian 2008 [75]	MTA* vs. DFC	MTA group (*n* = 30) DFC group (*n* = 30)	SSC	12 and 24 months	Absence of pain symptoms, tenderness to percussion, swelling, fistulation or pathologic mobility.	Absence of radicular radiolucency, internal or external root resorption, periodontal ligament space widening.	After 24 months 100% of DFC and MTA teeth were clinically successful.	After 24 months 100% of DFC teeth were radiographically successful. The radiographic follow-up evaluation revealed one failure (furcation involvement) in 18 molars treated with MTA after 24 months.	MTA could be used as a safe pulpotomy agent in cariously exposed primary molars and might be used as alternative to FC.
Agamy et al. 2004 [76]	Gray MTA (G-MTA) vs. White MTA (W-MTA) vs. FC (all not specified)	G-MTA group (*n* = 24) W-MTA group (*n* = 24) FC group (*n* = 24)	IRM and SSC	12 months	Absence of pain symptoms; tenderness to percussion; swelling; fistulation; pathologic mobility.	Absence of internal or external root resorption; periodontal ligament space widening.	At the 12-month evaluation, 100% of G-MTA teeth were clinically successful, while in the W-MTA group 3/18 showed clinical failure as well as two teeth in FC group.	At the 12-month evaluation, 100% of G-MTA teeth were radiographically successful, while in the W-MTA group 3/18 showed radiographical failure as well as two teeth in FC group.	In conclusion, G-MTA was superior to both W-MTA and FC as a pulp dressing agent for pulpotomized primary molars.
Eidelman et al. 2001 [77]	MTA vs. FC (both not specified)	MTA group (*n* = 30) FC group (*n* = 30)	SSC	30 months	Absence of pain; swelling; sinus tract.	Absence of internal root resorption; furcation radiolucency; periapical bone destruction.	MTA and FC showed 100% of clinical success. The follow-up evaluations revealed only one failure (internal resorption detected at 17-month postoperative evaluation) in a molar treated with FC.	MTA showed 100% of radiographical success. The evaluation of FC group, revealed only one failure (internal resorption). Pulp canal obliteration was observed in 9 of 32 (28%) evaluated molars. This finding was detected in 2/15 teeth treated with FC (13%) and in 7/17 treated with MTA (41%).	MTA showed promising clinical and radiographic success as a dressing material in the pulpotomy procedure of primary teeth.
Mettlach et al. 2013 [78]	Gray MTA* vs. DFC	MTA group (*n* = 119) FC group (*n* = 133)	IRM and SSC	12, 18, 24, 30, 36, 42 months	Authors stated that clinical success was scored based on modified scales adopted by Zurn and Seale.	Absence of pathologic nonperforated and perforated internal root resorption; external root resorption; inter-radicular or periapical bone destruction.	One tooth in the MTA group was judged to be a clinical failure (99% of success), and four teeth in the DFC group were judged to have failed (99% of success). There was no significant difference found between groups.	MTA group yielded a 95% of radiographical success, whereas DFC group showed 79%. This difference was found to be significant.	Gray MTA performed statistically better than DFC.
Durmus et al. 2014 [79]	DL vs. DFC vs. FS§§	DL group (*n* = 40) FC group (*n* = 40) FS group (*n* = 40)	GI and SCC	12 months	Absence of spontaneous pain, percussion/palpation, abscess, swelling, fistula, pathologic mobility.	Absence of periapical radiolucency, widened periodontal ligament space, pathologic internal/external root resorption, pathological changes of the alveolar bone in the furcation area.	After 12 months, a clinical success rate of 100%, 92.5% and 97% was observed in DL, FS and FC group, respectively. No statistically significant differences were detected between groups.	After 12 months, a radiographic success rate of 75%, 79% and 87% was observed in DL, FS and FC group, respectively. No statistically significant differences were detected between groups.	Pulpotomy performed with FS and FC provided comparable results. Although DL pulpotomy seemed to offer promising clinical success, it yielded low radiographic success rate.
Havale et al. 2013 [80]	FC### vs. GA vs. FS§§	FC group (*n* = 30) GA group (*n* = 30) FS (*n* = 30)	SSC	12 months	Absence of pain, tenderness, swelling, fistula formation, pathologic mobility.	Absence of widening of the periodontal ligament space, internal root resorption, external root resorption, pathological interradicular radiolucency, calcification of canal.	Clinical success was 96.7% for FS, 86.7% for FC and 100% for GA	Radiological success rates in FC, GA, and FS groups were 56.7%, 83.3% and 63.3%, respectively.	Although GA seemed the most efficient, FS and FC did not show statistically significant differences.
Huth et al. 2012 [81]	DFC vs., Er:YAG vs. CH†† vs. FS§§	FC group (*n* = 50) Er:YAG group (*n* = 50) CH group (*n* = 50) FS group (*n* = 50)	IRM and GI and SSC or composite resin restoration	12, 18, 24 and 36 months	Absence of spontaneous pain, tenderness to percussion, fistula, soft tissue swelling, pathological tooth mobility.	Absence of periapical or furcal radiolucency, pathologic external or distinct internal root resorption, widened periodontal ligament space.	After 36 months clinical success rates were: 92% for FC, 89% for Er:YAG, 75% for CH and 97% for FS.	Overall success after 36 months were: 72% for FC, 73% for Er:YAG, 46% for CH and 76% for FS.	After 36 months, CH was the least effective pulpotomy material, and FS was the most effective; however, FS did not show significant differences with FC. The Er:YAG laser showed comparable outcomes to FC.
Markovic et al. 2005 [82]	FS§§ vs. CH (not specified) vs. FC†††	FC group (*n* = 33) CH group (*n* = 34) FS group (*n* = 37)	GIC and amalgam	12 and 18 months	Absence of spontaneous pain, abnormal mobility, tenderness to percussion, fistula.	Absence of pathological changes of the alveolar bone in the apical and/or furcation area (visible periapical or inter-radicular radiolucency), integrity of lamina dura, pathological internal resorption, external root resorption.	The clinical success rate at 18 months for the FC and FS groups was 90.9% and 89.2% respectively. CH group showed an overall lower clinical success of 82.3%, although differences were not statistically significant.	RX success: FC 84.4%, CH 76.5%, and FS 81.1%. The differences between groups were not significant.	FS pulpotomy provided favorable clinical and radiographic success rates, comparable to FC pulpotomy. CH showed the worse performance among groups.
Ozmen et al. 2017 [83]	DFC vs. ABS vs. FS****	DFC group (*n* = 15) ABS group (*n* = 15) FS group (*n* = 15)	amalgam (in case of Class I cavities) or SCC (in case of Class II cavities)	24 months	Absence of spontaneous or severe pain, pathological mobility, swelling, sinus tract, tenderness to percussion, palpation.	Absence of furcal or periapical radiolucency, widened periodontal ligament spaces, internal or external root resorption, loss of lamina dura.	At the end of 24 months, the clinical success rates for ABS, DFC and FS were 87%, 87% and 100%, respectively.	RX success: DFC 80%, ABS 87%, FS 87%.	Comparable success was achieved using ABS, FC and FS as pulpotomy agents of deciduous teeth.
Farsi et al. 2015 [84]	NaOCl vs. DFC vs. FS (not specified)	NaOCl group (*n* = 27) DFC group (*n* = 27) FS group (*n* = 27)	ZOE and SCC cemented with GIC.	12 and 18 months	Absence of pain, swelling, sinus tract, mobility, pain on percussion.	Absence of internal root resorption, furcation radiolucency, periapical radiolucency, widening of the periodontal ligament space.	18 months: the clinical success rates were NaOCl 83.3%, FC 96%, FS 87%, respectively.	18 months: rx success rates were NaOCl 91.7%, FC 100%, FS 91.3%, respectively.	Comparable results were obtained using NaOCl, DFC and FS as pulpotomy agents for primary molars.
Jayam et al. 2014 [85]	white MTA** vs. FC°°°°	MTA group (*n* = 50) FC group (*n* = 50)	SCC and/or GI and amalgam.	24 months	Absence of history of pain, tenderness to palpation/percussion, pathological mobility, intra- or extra-oral swelling, intra- or extra-oral sinus.	Absence of integrity of lamina, radiolucencies in the apical or bifurcation areas of tooth, pathological internal or external root resorption.	MTA success rate was 100% in comparison to 90.48% success in FC group.	MTA success rate was 100% in comparison to 90.48% success in FC group.	MTA provided promising results as pulpotomy dressing material.
Srinivasan et al. 2011 [86]	MTA** vs. DFC	MTA group (*n* = 50) DFC group (*n* = 50)	SSC	12 months	Absence of spontaneous pain, draining fistula, swelling or abscess, mobility, premature exfoliation,	Absence of abnormal root resorption, internal root resorption, furcation involvement, periapical bone destruction.	After 12 months, DFC clinical success rate was 91.3%. In the MTA group, no clinical signs and symptoms were noted; thus, the clinical success was 100%.	Radiographic success rates were 78.26% and 95.74%, in DFC and MTA group, respectively.	MTA seemed to be clinically and radiographically superior to FC.
El Meligy et al. 2019 [87]	DFC vs. BD°	DFC group (*n* = 50) BD group (*n* = 50)	SSC	12 months	Absence of pain, swelling, tenderness to percussion, fistula, abnormal tooth mobility.	Absence of periodontal ligament space, periapical and furcation pathosis, internal resorption.	100% clinical success rates for both groups.	The BD group had a radiographic success rate of 100% at 12-month follow-up, while the DFC group had a success rate of 98.1% at 12 months.	BD and DFC pulpotomy techniques demonstrated favorable clinical and radiographic results in primary teeth, after a 12-month follow-up without any significant differences.
Sunitha et al. 2017 [88]	FC (not specified) vs. MTA* vs. EMP vs. PT	FC group (*n* = 50) MTA group (*n* = 50) EMP group (*n* = 50) PT group (*n* = 50)	SSC	12, 18 and 24 months	Absence of pain, swelling or abscess, sinus tract opening, mobility, pain on percussion.	Absence of pathological root resorption, widening of periodontal space, bifurcation radiolucency, and periapical radiolucency.	Clinical evaluation: FC 94%; PT 94%; MTA 100%; EMP 83%.	Rx success: FC 88%; PT 83%; MTA 94%; EMP 72%.	MTA was demonstrated to be a valid alternative to FC in pulpotomy procedures. PT and EMP were also proven to be promising agents.
Fernandes et al. 2015 [89]	CH (not specified) vs. DFC vs. LLLT vs. LLLT+ CH	CH group (*n* = 15) DFC group (*n* = 15) LLLT group (*n* = 15) LLLT+ CH group (*n* = 15)	IRM and GIC	12, and 18 months	Absence of spontaneous pain, mobility, swelling, fistula.	Absence of internal or external root resorption and furcation radiolucency.	All the groups were clinically successful over the follow-up period.	At 18 months follow-up, the radiographic success rate for the DFC group was 100%, 66.7% for CH group, 73.3% for LLLT group, and 75% for LLLT + CH group.	DFC provided the best results over the follow-up period. However, LLLT might be considered as an adjuvant alternative for vital pulp therapy on human primary teeth.
Subramaniam et al. 2009 [90]	MTA* vs. FC°°°°	MTA group (*n* = 20) FC group (*n* = 20)	SSC	24 months	Absence of pain, tenderness to percussion, gingival abscess, sinus/fistula, pathologic mobility.	Absence of internal root resorption, external root resorption, periapical/furcal radiolucency.	At the 12th month of evaluation a success rate of 95% and 85% was seen in the MTA and FC groups, respectively.	At the 12th month of evaluation a success rate of 95% and 85% was seen in the MTA and FC groups, respectively.	MTA provided highly promising results as pulpotomy agent.
Sonmez et al. 2008 [91]	DFC vs. FS (not specified) vs. CH (vs. MTA*	MTA group (*n* = 15) CH group (*n* = 15) DFC group (*n* = 15) FS group (*n* = 15)	amalgam (FS, DFC and CH groups) IRM and amalgam; (MTA group)	24 months	Absence of symptoms of pain, tenderness to percussion, swelling, fistulization, pathological mobility.	Absence of periradicular or interradicular radiolucency, internal or external root resorption, periodontal ligament space widening.	The success rates of CH (46.1%) and MTA (66.6%) were lower than FC (76.9%) and FS (73.3 %), although not statistically significant.	The success rates of CH (46.1%) and MTA (66.6%) were lower than FC (76.9%) and FS (73.3%), although not statistically significant.	CH appeared to clinically be less appropriate than FC, FS and MTA as pulpotomy dressing material.
Fuks et al. 1997 [92]	FC§§ and DFS	DFC group (*n* = 38) FS group (*n* = 58)	IRM and SSC	12-35 months (mean 20.5 months)	Absence of pain, swelling, sinus tract.	Absence of internal root resorption, furcation radiolucency, periapical bone destruction.	Total success rates of pulpotomies with FS and DFC were 92.7% and 83.8%, respectively.	Total success rates of pulpotomies with FS and DFC were 92.7% and 83.8%, respectively.	FS and DFC provided similar results.

ABS: Ankaferd blood stopper, Ankaferd Health Products Ltd.; BD: Biodentine; CH: Calcium hydroxide; DL: Diode Laser; DFC: diluted formocresol, 20% or one-fifth strength; Buckley’s Formocresol, Sulton Healthcare; EMP: enamel matrix protein, Emdogain, Straumann; FC: Formocresol; FS: Ferric sulphate; GA: glutaraldehyde, PSK Pharma, Karnataka; GI: glass-ionomer restorative material, KetacÔ Molar, Easy MixÔ, 3M ESPE; GIC: glass ionomer cement, Vitremer^®^, 3M ESPE; IRM: Reinforced zinc-eugenol cement, Dentsply.; LLLT: Low Level Laser Therapy; MTA: Mineral trioxide aggregate; NaOCl: Sodium hypochlorite; PT: Pulpotec, Products Dentaire – PD; SCC: stainless steel crown; ZOE: zinc-eugenol cement; * ProRoot MTA, Dentsply; ** Ângelus, Londrina; *** MTA-Plus, Avalon Biomed Inc; **** Hemospad, Spad Laboratorie; ° Biodentine, Septodont; °° Biodinâmica Química e Farmacêutica Ltd.a; °°° Sultan Chemists, Englewood; °°°° Pharmadent remedies Pvt. Limited; § Calcium enriched mixture cement (CEM), BioniqueDent; §§ Astringedent – Ultradent Products Inc; §§§ Cresol Formalinan, GHIMAS S.P.A; # Portland Cement (PC), Votorantim-Cimentos; ## Tempophore (TP), Septodont; ### Vishal Dentocare, Ahmedabad; † Biodinamica Quımica e Farmaceutica Ltd.a; †† Calxyl^®^, OCO Präparate GmbH; ††† Ja pan Dental Pharmaceuticals, Co. Ltd.

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
