# Peer review of "Different Pulp Dressing Materials for the Pulpotomy of Primary Teeth: A Systematic Review of the Literature"

_jcm, 2020, doi:10.3390/jcm9030838_

Round 1
Reviewer 1 Report
This is a very interesting review about different pulp dressing materials for the pulpotomy of primary teeth. The manuscript is well-written and organized, some minor revisions of the English language are recommended.
Two aspects that the Authors should address are:
- Why authors included also studies n° 79, 81, 84 and 89 among the 41 papers reported in Table 3? In lines 340/343 the authors declare that these studies were excluded in the evaluation of pulp dressing agents.
- It would be interesting to understand the influence of restoration material after the use of FC or FS as pulp fixative materials. Are there some studies investigating about this topic? If not it should be reported that it might influence the result over time.
Author Response
REVIEWER 1
This is a very interesting review about different pulp dressing materials for the pulpotomy of primary teeth. The manuscript is well-written and organized, some minor revisions of the English language are recommended.
The Authors are grateful for the Reviewer’s considerations.
Two aspects that the Authors should address are:
- Why authors included also studies n° 79, 81, 84 and 89 among the 41 papers reported in Table 3? In lines 340/343 the authors declare that these studies were excluded in the evaluation of pulp dressing agents.
The aforementioned studies had been included in the review since they reported on comparison between different pulp dressing materials. However, the confounding factors such as such as diode laser (ref #79), Er:YAG laser (ref # 81), sodium hypochlorite (ref #84), and low level laser therapy (ref #89) were not considered to avoid misinterpretation of the results due to the high variability given by devices or materials different from the evaluated ones.
Specifically, in the study #79 conducted by Durmus et al. (2014) just data obtained by the comparison between FC and FS were considered. In the study #81 (Huth et al. 2012), pulpotomized primary molars treated with CH, FS and FC were compared, excluding molars treated with Er:YAG laser. In the study #84 by Farsi et al. (2015), pulpotomies performed with FC and FS were evaluated, avoiding to consider the ones obtained with sodium hypochlorite. Finally, in the study #89 (Fernandes et al. 2015) data provided by the treatment with FC and CH were analyzed, excluding the outcomes belonging to low laser therapy group.
- It would be interesting to understand the influence of restoration material after the use of FC or FS as pulp fixative materials. Are there some studies investigating about this topic? If not it should be reported that it might influence the result over time.
This topic is of central interest in the maintaining of the primary teeth over time after pulpotomy procedures. Due to this reason, the Authors had considered as inclusion criteria the presence of “Definitive restorations of the primary teeth”. However, as stated in the manuscript, the restoration materials reported by the included studies were different (composite, amalgam, glass ionomer cement, stainless steel crowns), although definitive. To the best of our knowledge, it’s not been established yet the influence of different restoration materials on the success of pulpotomy over time. Only the presence of eugenol (contained in the temporary restoration) had been reported to alter the adhesion properties of the subsequent composite restoration; however, according to the data provided by the included studies, it’s impossible to drawn an univocal conclusion in these terms. Moreover, it should be noticed that the time between the pulpotomy treatment and the physiological exfoliation of the same tooth may be widely variable, rendering very hard to follow the procedure success/fail in the middle and long term.
The Authors are in agreement with the Reviewer’s concerns, therefore, the aforementioned consideration have been added within the text (page 11, lines 347-350).
Reviewer 2 Report
Article appears sound.
Minor edit for page , line ;
either change "difference" to "different or change to difference of" composition.
Also, the numbers less than 10 should be spelled out unless they are apart of an interval.
Author Response
REVIEWER 2
Article appears sound.
Thank you.
Minor edit for page, line ;
either change "difference" to "different or change to difference of" composition.
The suggested change has been undertaken at page 6 line 168 as follows: “…and might have a slightly different composition”.
Also, the numbers less than 10 should be spelled out unless they are apart of an interval.
Numbers less than 10 have been spelled out within the whole manuscript, as requested.
Reviewer 3 Report
The aim of the present systematic review was to compare different materials for pulpotomy in primary teeth. The topic is very interesting and important. However, some minor points should be addressed:
- If the quality of the included studies was assessed by two or more independent authors, what was the inter-examiner agreement?
- References - Please check the numbering
Author Response
REVIEWER 3
The aim of the present systematic review was to compare different materials for pulpotomy in primary teeth. The topic is very interesting and important. However, some minor points should be addressed:
If the quality of the included studies was assessed by two or more independent authors, what was the inter-examiner agreement?
The requested additional information have been reported within the Materials and Methods section (page 4, lines138-139) as well as Results section (page 6, lines 184-185), explaining how the quality assessment of the included studies was conducted and the relative inter-examiner agreement.
References - Please check the numbering
References have been wholly checked and correctly renumbered, as requested.